# Review of the Standard and Advanced Screening, Staging Systems and Treatment Modalities for Cervical Cancer

**DOI:** 10.3390/cancers14122913

**Published:** 2022-06-13

**Authors:** Siaw Shi Boon, Ho Yin Luk, Chuanyun Xiao, Zigui Chen, Paul Kay Sheung Chan

**Affiliations:** Department of Microbiology, The Chinese University of Hong Kong, Shatin, N.T., Hong Kong SAR, China; boonss@cuhk.edu.hk (S.S.B.); peterlukhoyin@link.cuhk.edu.hk (H.Y.L.); chuanyunxiao@cuhk.edu.hk (C.X.); zigui.chen@cuhk.edu.hk (Z.C.)

**Keywords:** cervical carcinoma, human papillomavirus, cervical cytology, HPV genotyping, cervical cancer staging, cervical cancer treatment

## Abstract

**Simple Summary:**

This review discusses the timeline and development of the recommended screening tests, diagnosis system, and therapeutics implemented in clinics for precancer and cancer of the uterine cervix. The incorporation of the latest automation, machine learning modules, and state-of-the-art technologies into these aspects are also discussed.

**Abstract:**

Cancer arising from the uterine cervix is the fourth most common cause of cancer death among women worldwide. Almost 90% of cervical cancer mortality has occurred in low- and middle-income countries. One of the major aetiologies contributing to cervical cancer is the persistent infection by the cancer-causing types of the human papillomavirus. The disease is preventable if the premalignant lesion is detected early and managed effectively. In this review, we outlined the standard guidelines that have been introduced and implemented worldwide for decades, including the cytology, the HPV detection and genotyping, and the immunostaining of surrogate markers. In addition, the staging system used to classify the premalignancy and malignancy of the uterine cervix, as well as the safety and efficacy of the various treatment modalities in clinical trials for cervical cancers, are also discussed. In this millennial world, the advancements in computer-aided technology, including robotic modules and artificial intelligence (AI), are also incorporated into the screening, diagnostic, and treatment platforms. These innovations reduce the dependence on specialists and technologists, as well as the work burden and time incurred for sample processing. However, concerns over the practicality of these advancements remain, due to the high cost, lack of flexibility, and the judgment of a trained professional that is currently not replaceable by a machine.

## 1. Introduction

Cervical cancer is a disease where cells in the uterine cervix, a region that connects the vaginal and upper uterus, grow uncontrollably. According to the latest Global Cancer Statistic 2020 (GLOBOCAN), cervical cancer continues to be the fourth most common cancer worldwide, with 604,127 estimated new cases and 341,831 deaths across the globe [1]. One of the important, though in itself insufficient, aetiological agents contributing to the malignancy of the cervix uteri is the human papillomavirus (HPV) infection [2]. Nonetheless, persistent HPV infection is an essential factor contributing to over 99% of cervical cancer cases [3]. Among the over 200 HPV genotypes identified, the 13 so-called “high-risk” (HR-HPV) HPV genotypes, including HPV16, 18, 31, 33, 35, 39, 45, 51, 52, 56, 58, 59, and 68, are highly associated with the cervical cancer risk, while the “low-risk” HPV (LR-HPV) genotypes 6, 11, 40, 42, 43, 44, 53, 54, 61, 72, and 81 often cause benign lesions. Due to the accountability of HPV16 and 18 for over 70% of cervical cancer cases, HPV screening programs prioritise the detection of these 2 HPV genotypes [4,5,6]. While there is a dramatic declining trend in the incidence and mortality of cervical cancer reported in high-income countries [7], unfortunately almost 90% of both the incident cases and the mortality is reported from low- and middle-income countries, such as those in sub-Saharan Africa, Melanesia, South America, and Southeast Asia [1,8].

In general, the World Health Organization (WHO) has defined premalignant lesions of the uterine cervix as cervical intraepithelial neoplasia (CIN) [9], while the epithelial tumours of the uterine cervix are histologically classified into squamous cell carcinoma, glandular tumours and precursors, mesenchymal tumours and tumour-like conditions, mixed epithelial and mesenchymal tumours, melanocytic tumours, miscellaneous tumours, lymphoid and haematopoietic tumours, and secondary tumours [10]. Among these, squamous cell carcinoma (75%) is the most commonly reported cervical cancer type, followed by adenocarcinoma, a type of the glandular tumour and precursors (25%) classification [11,12]. HPV16 is predominantly found in squamous cell carcinoma, while HPV18 is more commonly detected in adenocarcinomas and adenosquamous carcinoma [13]. For these two types of cervical cancers, the risk factors, the HPV genotypes detected, and the treatment are similar [14], while the usual endocervical adenocarcinoma types, such as the mucinous, micropapillary, and villoglandular, are associated with HPV, and the gastric, clear cell, endometrioid, and mesonephric types are often not associated with HPV [14,15].

## 2. From HPV Infection to Precancerous Formation

Virtually all of us are susceptible to HPV infection once in a lifetime, regardless of gender, genetic background, and geographical location. The virus is present ubiquitously in the environment. Upon gaining access to the basal cell of the squamous epithelium of the cervix uteri of a woman through microlesions, the virus enters the cells and establishes the viral life cycle. Once the virus gains entry, which is believed to occur via the clathrin or caveolin-independent endocytosis mechanism, the virus traffics through the subcellular compartments, uncoats, and translocates viral DNA into the nucleus. This then allows the virus to deploy the host cell replication machinery and start viral genomic replication [16]. The expression of viral proteins in the basal layer stimulates cell proliferation and causes dysregulation of the normal cell cycle [17], hijacking the terminal differentiation and overtaking the host replication machinery to force cell cycle re-entry for the viral genome amplification and packaging. Hence, a thicker suprabasal layer is generated. When the HPV-infected cells reach the upper layer of the epithelium, the virions are shed, and then they release the virus to conjugated cells with squamous flakes [18]. The infectious virions can be transmitted through sexual intercourse or self-inoculation [19]. Usually, genital HPV infections are transient, and most viruses will be cleared out without developing into cancer. However, recurrent or persistent HPV infection will lead to the transformation of keratinocytes into intraepithelial neoplastic cells, which may develop into malignancy [20].

The implementation of cytology tests, HPV screening, and HPV vaccination programs has successfully reduced the cervical cancer burden, particularly in developed countries [7]. In the following sections, the standard and state-of-the-art diagnostic methods, the precancer and cancer classification systems, as well as the clinical management and treatment modalities used in clinical trials, are discussed.

## 3. Detection of Premalignancy and Malignancy of the Uterine Cervix

As early as the 1920s, techniques had been developed to scrutinize vaginal and cervical cytological samples for the detection of cancerous abnormalities. In the 1940s, a Greek physician, Georgios Nikolaou Papanikolaou, introduced the Papanicolaou stain to detect abnormal precancerous and cancerous morphological changes in cervical cytology samples, which is still widely used today. Other available detection methods include visual inspection, and detection of HPV nucleic acids and biomarkers predictive of cervical cancer. In the 2020s, with the advancement of technology, artificial intelligence (AI) has been a popular module incorporated for detecting and reading cervical cytology samples [21,22,23,24,25,26,27,28,29,30,31,32,33]. The use of AI techniques can alleviate the scarcity of professional resources and the heavy workloads, and at the same time increase the accuracy and specificity of diagnosis. The timeline of the development of cervical cancer detection methods is shown in Figure 1.

### 3.1. Standard Detection Methods

A systematic screening program for the detection of neoplastic cervical cells is undoubtedly an effective measure to decrease the incidence and mortality of cervical cancer. Different countries may adopt slightly different screening strategies. This may depend on the availability of manpower, the health infrastructure, and the resources. The WHO published a standard screening method in the WHO HPV Laboratory Manual [34], which includes visual inspection, the Pap test, HPV nucleic acids detection, and genotyping.

#### 3.1.1. Visual Inspection

Visual screening techniques can be classified into low- and high-technology approaches. The low-technology approach includes direct visual inspection (DVI), while the high-technology approach includes those that utilise electro-optical detectors to identify cervical cancer precursors and invasive cervical cancer. When performing DVI, 3–5% of acetic acid was applied to the uterine cervix of patients, and then, the inspection was performed using either the naked eye or a low-power magnifying device to identify the cervical lesion. Unfortunately, this method proved to be relatively nonspecific, and many women who lacked significant cervical lesions were classified as positive on the DVI [35]. In the 1980s, visual screening had a renewed interest as the WHO had introduced the concept of “downstaging” as a low-cost alternative screening method [36]. Thereafter, massive visual detection methods have been developed. Speculoscopy is a variant of DVI, in which a special chemiluminescent light is used to illuminate the cervix and the cervix is viewed with a magnifying device after the application of 5% acetic acid [37]. Cervicography is another visual screening method, which was developed by Adolf Stafl in 1981 [38]. This method is more sensitive in detecting squamous intraepithelial lesions (SIL) than cytology, but its specificity is much lower [39]. Meanwhile, colposcopy refers to the direct observation of cervical epithelial lesions or vaginal lesions with a strong light source and colposcope, which can effectively improve the accuracy of the clinicians’ judgment of cervical lesions. In addition, a low-cost, high-resolution microendoscope (HRME) was developed and used for the direct visualization of neoplastic biomarkers during colposcopy [40].

#### 3.1.2. Cervical Cytology Detection

A Pap smear of the cervix is a simple, quick, and inexpensive screening procedure to detect cytological changes in the uterine cervix. This has also been the most widely adopted method worldwide since the 1940s [41]. Over the past 80 years, the cervical cancers and precancers of countless women have been detected through the Pap smear, resulting in timely treatment [42]. However, the manual analysis of Pap smear samples is time-consuming, laborious, and error-prone. To tackle these shortcomings, liquid-based cytology tests, were developed. Liquid-based cytology tests, including ThinPrep and SurePath, have largely replaced conventional Pap tests for cervical cytology screening due to their higher sensitivity in comparison to that of the conventional method [43]. The ThinPrep technique uses a methanol-based PreservCyt as a fixative liquid. It also uses a vacuum for cell dispersion and collection, followed by the use of air pressure to create the cytology slides. Unlike ThinPrep, SurePath uses an ethanol-based preservative fluid. This method uses centrifugation and resuspension of the cells in a sucrose density gradient. Gravity is used to transfer the cells to the slide [44]. However, due to the high technical demand, as well as the costly consumables and equipment, the implementation of the liquid-based cytology test remains challenging in low- and middle-income countries.

Alternatively, DNA ploidy analysis and TrueScreen can be performed for the detection of cervical hyperplasia and dysplasia. These methods are preferred due to their low cost, non-dependence on cytopathologists, and they can be applied in a large-scale population screening [45]. DNA ploidy analysis is performed to assess the physiological state and the pathological changes of cells by measuring the DNA content or chromosome multiplication in the nucleus. Meanwhile, TrueScreen is a real-time instrument for the detection of precancerous lesions or cancer of the uterine cervix. Its working principle is based on the photoelectric physiological basis of biological tissue. It compares the photoelectric information that Atlas collected from the patient samples with the digital standard histopathological database stored in the instrument body and simulates the analysis process of pathological film readers. An added benefit is that the test results can be printed immediately [46]. Therefore, some people believe that TrueScreen can replace the liquid-based cytology test as a means of cervical cancer screening in areas with relatively poor medical resources, in underdeveloped economies, grass-roots community hospitals, and physical examination centers. In economically developed areas, TrueScreen can also be combined with other examinations to reduce the chance of a missed diagnosis and to increase the sensitivity of screening [47,48]. Even though the Pap test, liquid-based cytology, and other cytology tests allow the detection of premalignancy or malignancy of the uterine cervix [49], these tests do not specifically discriminate as to whether or not the morphological changes are contributed by HPV infection.

#### 3.1.3. Human Papillomavirus (HPV) Nucleic Acid Detection and Genotyping

At the beginning of the 1980s, zur Hausen and his team identified the HPV genotype 16 as an important aetiological agent contributing to genital cancer and revealed the genetic organization of HPV DNA in cervical cancer cells [50]. In 1986, it was first reported that the expression of the HPV16 oncogenes, E6 and E7, can be used as the tumour markers of cervical cancer. The E6 and E7 transcripts could provide a sensitive, early predictor of the cervical cancer risk in women with normal and minor cytological alteration [51]. In addition to detecting the intralesional HPV transcripts, the HPV circulating tumour DNA (HPV ctDNA) was proposed as a biomarker for the detection and disease monitoring of HPV-related cancers. One study suggests that the HPV16 ctDNA biomarker appeared to be highly specific; however, it lacks the sensitivity for the detection of cervical cancer, even for those at an advanced tumour stage [52].

The polymerase chain reaction (PCR)-based assay is a promising method for the detection of HPV DNA [53]. The PCR is highly sensitive for the detection of the viral nucleic acid shed from the uterine cervix into the vaginal canal [54] and precisely differentiates the HPV genotypes present in the samples. The test is sensitive for samples collected by a health practitioner or oneself. Self-sampling has become popular and acceptable among women as this method is convenient, flexible, and can be performed in a variety of settings, while avoiding the need to perform a pelvic examination and the awkwardness, particularly among women in a conservative society [54]. More importantly, the HPV detection rate of self-collected samples is comparable to that performed by health practitioners [55]. HPV genotyping is helpful to further triage primary HPV, co-test positive women, and determine the following management steps.

Currently, there are FDA-approved and validated HPV tests, including the Hybrid Capture 2 (HC2) assay (Qiagen, Gaithersburg, MD, USA), based on the DNA-Probe-Hybrid immunoassay technique; CervistaHPV HR and Cervista™ (Hologic, San Diego, CA, USA) HPV16/18, based on DNA-probe technology; the Cobas 4800 HPV test (Roche Molecular Systems Inc., Pleasanton, CA, USA) and the PCR-based BD Onclarity™ HPV assay (BD Diagnostics, Sparks, MD, USA); the APTIMA HPV assay (Hologic, San Diego, CA, USA), based on RNA capture and the amplification of HPV RNA [56]; and the Luminex Genotyping GP HR (Diassay, The Netherlands) immunoassay, as well as the reverse-hybridization-based INNO-LiPA HPV Genotyping (Fujirebio, Gent, Belgium) [57]. The target regions for these tests are the L1, E6, or E7 genes encoded by HPV, with a sensitivity of about 80% [58]. While these qualitative tests allow the detection of HPV genotypes in samples, advancement has made possible the inclusion of the use of the automated platform. For example, Xpert HPV (Cepheid, CA, USA) is a quantitative automated platform with sample processing, cell lysis, real-time PCR, and detection performed in a self-contained cartridge. While the Linear Array (Roche Molecular Systems Inc., CA, USA) genotyping test shows that there is significant consistency between the detection of any HR-HPV genotype and the E1 determination based on quantitative polymerase chain reaction (qPCR) [59]. Furthermore, the next-generation sequencing (NGS) can be explicitly used to improve the specificity and sensitivity of HPV detection. The NGS panel can effectively detect the presence of all HPV genotypes present in archival formalin-fixed, paraffin-embedded (FFPE), liquid-based cytology (LBC), and plasma samples [60,61]. Thus, the NGS technology lowers the chance of false-negative results, compared with the traditional PCR-based assays, and provides more accurate screening results for the subsequent diagnosis and treatment planning.

#### 3.1.4. Viral and Cellular Biomarkers

In addition to HPV nucleic acid and cellular morphological changes, the phenotypic cancer biomarker provides valuable evidence of protein expression alteration in the cervical biopsy. The immunostaining of tissue for p16^INK4A^, Ki-67, and p53, are often included as surrogate markers for HPV-associated cervical cancer. The dual staining of these biomarkers, for example, p16 and p53 immunohistochemistry test have been commonly performed in many laboratories worldwide. The overexpression of p16^INK4A^ is indicative of a negative feedback loop following the degradation of retinoblastoma (pRB), a key cell cycle regulator, by HPV-encoded E7 [62]. Ki-67 is a nuclear protein associated with cell proliferation, while p53, which is also a tumour suppressor and a cell cycle regulator, is a prominent degradation target of HPV-encoded E6 [63]. Other biomarkers suggested to be included in the early evaluation for cervical cancer screening include Cdc6 and MCM5 for cell proliferation [64,65], and cytokeratin 19 mRNA for lymph node metastasis [66], as well as mir124-2, FAM19A4, and SEPT9 for DNA hypermethylation [67,68,69]. These markers have clinical significance and are helpful in disease prevention and management.

### 3.2. AI-Assisted Cervical Cancer Detection

With the rapid advancement of present-day clinical and computer technology, computer-aided (CAD) architectures are integrated into various screening and diagnostic strategies. As an auxiliary tool, artificial intelligence (AI) technology is also widely used in the field of medical diagnosis. The increasing interest in the development of CAD diagnosis systems for cervical cancer screening is closely related to the common practical difficulties experienced in under-resourced health facilities, with the shortage of trained cytotechnologists and equipment. Computer vision and machine learning approaches are often used in CAD systems to reduce the dependence on manual microscopic examination of cervical cytology smears. The CAD systems can be applied to the automated handling of smear variability, the detection of artifacts, the segmentation of individual cells and cell clusters [70,71], the segmentation of nuclei and cytoplasm for each cell [72], the automated detection of atypical cells or abnormal changes in cell morphology [71,73], and the accurate classification of the cervical cells [70,74]. These technical models, combined with other data sets or with clinical data, have greatly expanded the scope of application. For example, various Feature Selection Technique (FST) methods were applied to the transformed datasets to predict cervical cancer or identify important risk factors [75,76]. The machine learning algorithm is fused with an optoelectronic sensor to realise rapid sample measurement and the automatic classification of results [77]. Several meta-analyses confirmed the diagnostic performance of machine learning or deep learning algorithms in cervical cancer recognition [78,79]. In addition, the integration of AI technology in analysing Pap smear samples reduced the processing time drastically to a few seconds, without compromising the sensitivity when compared to the referenced standard produced by pathologists [80]. In addition, the studies also showed that AI-assisted technology has a comparable or even better ability for detecting >90% CIN2 and CIN3 in a liquid-based cytology test [81] or colposcopy [82] samples than manual reading by a trained specialist.

## 4. Staging System for Cervical Intraepithelial Neoplasia and Cancer of Uterine Cervix

Through physical examination, visual inspection, and cytologic and histologic screening, the premalignant and malignant tissues are differentiated and classified primarily using two systems—the cervical intraepithelial neoplasia (CIN) and the FIGO system. The CIN system is used to classify the thickness of the precancerous lesion, while the FIGO system is used to stage malignant tissue based on the size of the tumour, the involvement of the lymph nodes, and the spread of the cancer cells. The latter system often liaises with the tumour, node, and metastasis (TNM) system.

### 4.1. Cervical Intraepithelial Neoplasia (CIN) System

The cervical intraepithelial neoplasia (CIN) system is a classification system that evolved in 1968 to classify precancerous lesions of the cervix uteri based on the degrees of dysplasia. Cervical biopsies and histological examinations are used to classify CIN into 3 degrees—CIN1 (mild), CIN2 (moderate), and CIN3 (severe dysplasia) [83,84], as illustrated in Figure 2. CIN1 involves dysplasia in the lower third or less of the epithelium, while dysplastic cells are observed covering most of the epithelium layers in CIN2. In CIN3, dysplasia progresses to the entire thickness of the epithelium. Dysplasia becomes cancer once it invades through the basal membrane [85]. In 1988, the Bethesda system was introduced in cytological reports, in which precancerous lesions are classified into two categories, namely low-grade (LSIL) and high-grade squamous intraepithelial lesion (HSIL). CIN1 is equivalent to LSIL. Many cases of CIN1 or LSIL may not be related to high-risk HPV genotypes. Considering the difficulty in distinguishing the cytological appearance between CIN2 and CIN3, the Bethesda system combined these two groups into the HSIL [86].

### 4.2. FIGO System

The International Federation of Gynecology and Obstetrics (FIGO) system is a staging system for gynaecologic cancers. The system was established in 1958 to recognise the recurrence rate and patients’ outcomes with regard to the degree of tumour spread from initial sites. In response to the increased understanding of cancer, the system was then consistently revised and updated in 1988, 2009, 2014, and 2018 [25,87]. This clinical staging system has been predominantly used in clinical practice and reporting worldwide.

In the past, cervical cancer was mostly determined preoperatively, based on clinical examination. This was mainly due to the high prevalence of cervical cancer, particularly in developing countries, due to the lack of advanced imaging systems and uniform staging guidelines. The earlier FIGO system was based on pelvic examination, lesion biopsy, radiography, colposcopy, cystoscopy, sigmoidoscopy, barium enema (BE), and intravenous pyelogram (IVP). This staging was rather inaccurate as compared to surgical staging, in which 20% to 40% of stage IB—IIIB cancers were under-staged, and >60% of stage IIIB cancers were over-staged [88]. Misestimating the primary stage will affect the judgement of prognostic factors, and it is extremely important to have a precise judgement on cancer staging. Hence, in the latest 2018 FIGO staging system, additional imaging and pathological findings are taken into consideration when assigning the cancer stage [25]. Stage IB was included in the latest version. Based on screenings, either from the Pap test and/or the HR-HPV test, in the case of any indication of cervical cancer clinicians may conduct further examination using colposcopy or conization to obtain the cervical biopsy [89]. If malignant cells are found from biopsy, the patient may need to take cross-sectional and functional imaging, such as X-ray, computed tomography (CT,) positron emission tomography (PET), and magnetic resonance imaging (MRI) for accurate disease assessment and staging [90,91]. The latest FIGO staging is shown in Table 1 (modified from the 2018 FIGO Staging System for Uterine Cervical Cancer [92]).

### 4.3. TNM System

The American Joint Committee on Cancer (AJCC) provided a guideline for the tumour, node, and metastasis (TNM) staging system (Table 1). The latest 2019 version is the version 9 AJCC TNM cervical cancer staging. The old versions only allow plain radiographs, such as intravenous pyelography, while the updated version allows cross-sectional imaging to be incorporated into staging [93]. The details of each TNM category are discussed below.

#### 4.3.1. Tumour (T) Category

The T category is used to describe the size of the primary tumour and the depth of tumour invasion into adjacent tissues, which is staged numerically. TX represents that the primary tumour cannot be assessed, while T0 indicates there is no evidence of a primary tumour. T1–4 represent the tumour size and the tumour invasion, as shown in Table 1. As with the FIGO system, tumour size can be determined using physical examination, pathological evaluation, and imaging studies [91,93].

#### 4.3.2. Lymph Node (N) Category

The N category is used to describe the involvement of regional lymph nodes in the tumour. N0 indicates no lymph node metastasis, while N1–2 indicate the degree of nodal spread. N1 indicates the pelvic lymph node involvement, and N2 indicates para-aortic lymph node involvement. The detailed N staging is described in Table 2.

The involvement of the lymph node is one of the most important prognostic factors for cervical cancer. Lymph node metastasis is often associated with poor prognosis and its involvement is critically considered in cancer treatment. However, it is challenging to diagnose lymph node metastasis using preoperative examination [94]. Open radical hysterectomy with pelvic lymph node dissection (ORH) was the standard method for diagnosis and surgical treatment in the past century. After ORH, patients tend to suffer from massive blood loss, bladder dysfunction, and blood-transfusion complications. The use of laparoscopy can circumvent this; this widely used technique provides minimal invasion and reduces pain and blood loss; less blood transfusion is required, and the hospital stay is shortened [95,96].

#### 4.3.3. Metastasis (M) Category

M category defines the presence of distant metastases of the primary tumour, and the latest description is shown in Table 3 [97]. Such metastases can be detected using physical examination, cross-sectional imaging, core-needle biopsy, fine-needle aspiration, incisional biopsy, excisional biopsy, and surgical resections [93]. The most prevalent organ where cervical cancer cells spread is the lung (37.9%), followed by bone (16.7%) and liver (12.5%). While patients of an age older than 60 are more likely to develop multi-site metastasis, involving mainly lung and liver (9.1%), followed by lung and bone (8%) and three of these sites (lung, liver, and bone) (5.8%) [98]. Ageing is also a key prognostic factor, with emerging data suggesting that age-related changes in the tumour microenvironment, in immunosenescence, and in a lower treatment–response rate will promote tumour metastasis. Age-related senescence induces fibroblasts to release the expression of senescence-associated secretory phenotype (SASP). SASP includes signalling factors, such as chemokine and interleukin proteases to promote inflammation that damages the surrounding tissues. Immunosenescence accumulates with ageing, the declined immune function allows tumour cells to evade immune surveillance, which fosters metastatic dissemination [99,100]. In rare cases, the cervical tumour can spread to the brain, heart, and muscles [101].

## 5. Cervical Cancer and Treatment

The premalignancy of the uterine cervix is preventable and treatable if neoplasia is detected early. “Screen-and-treat” is a commonly adopted clinical management for precancerous lesions. In general, the standard curative options for precancers include large loop excision of the transformation zone (LLETZ) or loop electrosurgical excision procedure (LEEP), cryotherapy, and cold knife conization, while for locally advanced cervical cancer, hysterectomy, radiotherapy, chemotherapy, and radiotherapy with concurrent chemotherapy and immunotherapy are offered to the patients [102].

### 5.1. Screen-and-Treat Strategies

The screen-and-treat approach is mainly recommended for women of age ≥ 30, due to their higher cervical cancer risk than younger women. The screen-and-treat strategy is summarised in Figure 3. According to the guidelines published by the WHO [103], the HPV nucleic acid detection and visual inspection with acetic acid (VIA) are the primary screening tests. It is strongly recommended that women of age 30 to 49 years old should have regular screening at 5- to 10-year intervals. While for women aged 50 and above, after receiving the consecutive negative cervical cancer screening results, further screenings is not necessary.

To perform an HPV nucleic acid detection test, the samples can either be self-sampled or collected by a health practitioner. In the case when HPV DNA is detected, depending on the eligibility, women should be assessed through visual inspection with acetic acid (VIA). VIA at the transformation zone is recommended for pre-menopausal women. This is because the transformation zone starts to recede during menopause. Ablative treatment can be performed via cryotherapy, in which the abnormal lesion is removed by heating it with thermal coagulation. Meanwhile, for high-grade lesions (CIN 2 or 3), cryotherapy is the first choice of treatment, provided that the entire lesion and the squamous-columnar junction are visible, and the lesion does not cover 75% of the ectocervix region. In the case when cryotherapy is not recommended, excisional treatment via loop electrosurgical excision procedure (LEEP) can be performed [102]. For women who are ineligible to perform ablative treatment, LEEP or cold knife conization should be performed, followed by histological assessment for lesion classification. The patients should attend follow-up within a year after the treatment.

For patients diagnosed with precancer or malignant cervical cancer, surgery is the main therapeutic measure. For the treatment of cervical cancer, the National Comprehensive Cancer Network (NCCN) [104] and the European Society for Medical Oncology (EMSO) [105] published comprehensive guidelines. For advanced, invasive, and metastatic cervical cancer, the tumour is usually removed by performing hysterectomy (simple or radical), trachelectomy [106], and pelvic exenteration [107], depending on the size and nature of the tumour. Often, when performing a radical hysterectomy, para-aortic lymph node sampling, pelvic lymphadenectomy, or sentinel lymph node mapping and biopsy is conducted [106]. It is crucial to know the involvement of lymph nodes for the planning of the subsequent adjuvant therapy.

In addition to surgery, for patients at the advanced stages, with either locally advanced carcinoma with or without lymph node involvement, or invasive or metastatic carcinoma, radiotherapy, cisplatin or cisplatin-based chemotherapy, radiotherapy concurrent chemotherapy and immunotherapy are offered. These therapies ultimately sensitize the cancer cells to undergo apoptosis, and the dead cells are subsequently eliminated by the immune system. These treatment modalities have been widely applied as standard treatments; however, they lack tissue selectivity and specificity. In addition, most of them are delivered systemically. This renders high toxicity and poses adverse consequences affecting multiple organs. Several onco-target compounds have been designed aiming at features of cancer. As cancer cells are often surrounded by blood vessels, anti-angiogenic drugs are designed to obstruct the infrastructure that provides nutrients to the tumour site. Another approach is to use a drug that has a tissue-targeting specificity and inhibits the oncogenic signalling pathways. In the following sections, we discuss the safety and efficacy of different treatment modalities in trials to treat cervical cancer patients, including the advancement of laparoscopy, radiotherapy, chemotherapy, immunotherapy, anti-angiogenic and target-specific drugs, drug-antibody conjugates, and HPV vaccines.

#### 5.1.1. Robotic-Assisted Laparoscopy

Laparoscopic surgery is a type of minimally invasive surgery (MIS) that is less invasive than conventional open surgery. Laparoscopy is performed through a small incision (0.5–1.5 cm), where a small surgical instrument, light source, and a camera are inserted into the abdomen/pelvis, for diagnosis and/or surgery. In the 1970s, the concept of surgical robots was endorsed by the National Aeronautics and Space Administration (NASA) to replace the physical presence of surgeons in space or military zones. Since then, surgical robots have been developed and implemented for different types of surgery, such as, for example, orthopaedics (unicompartmental knee replacement [108]), ophthalmic (retinal vessel cannulation, membrane peeling [109]), neurosurgery (brain tumour removal [110], deep brain stimulation [111]), thoracic (vascular resection [112]), hepatobiliary (liver resection [113]) and robot-assisted laparoscopy in gastric, pancreatic, urology, and gynaecology surgery [114].

The research and development of surgical robots have been increasingly active in recent decades. A robot is a device combining mechanics, electronics, and information that can be controlled manually or programmed to perform specific tasks. Surgical robots can be divided into master-slave robots or hand-held robotic forceps, which were developed to fit different surgical procedures. A master-slave surgical robot usually has a 6-degrees-of-freedom (DOF) motion. Surgeons can operate 4-DOF arms outside the abdominal cavity and a 2-DOF wrist joint at the tip. They can operate the remote slave arms directly in the master console or perform telesurgery through a network. One major disadvantage of the master-slave robot is that the master console requires large space and high installation and operating costs [114,115]. Hence, hand-held robot forceps were developed. For example, the Kymerax© Precision-Drive Articulating Surgical System was developed by Terumo© Medical Corporation. This instrument offers 6-DOFs and a wrist joint tip controlled by digital buttons in the handle, which is connected to the main console by cables [116]. JAiMY©, developed by Endocontrol Medical, provides the smallest (5 mm) robotic needle. This instrument has two intracorporeal DOF, controlled by a joystick connected to an ergonomic handle. The design could resolve fatigue due to long surgery [117].

In the early 2000s, the U.S. FDA gave clearance for the marketing of a robotic device, the da Vinci Robotic System (dVRS), for laparoscopic surgery. The advanced model, da Vinci Xi also gained FDA approval shortly after the success of the dVRS. Nonetheless, laparoscopy may face several limitations, such as limited range of motion and vision, surgeon fatigue, and ergonomic restrictions. With the continuous advancement of technology, these shortfalls can be overcome [118]. In 2019, the FDA issued a warning over the use of robotic-associated surgical devices in cancer-related surgeries. When comparing the clinical outcomes between the use of robotic-assisted surgical devices and open abdominal surgery or MIS, the rate of recurrence and motility did not differ. However, MIS was associated with a lower rate of long-term survival compared to open surgery [119]. The ergonomic design of the instrument can be improved; however, the installation and operating costs remain a major concern. The average cost for robotic-assisted laparoscopy is significantly costlier than laparoscopic surgery, which was estimated to be USD 12,340 ± 5880 and USD 10,227 ± 4986, respectively. This higher cost is predominantly related to the operating procedure [120]. Despite the concern over the cost-effectiveness of robotic-assisted surgery in cancer treatment, the FDA authorized the Hominis Surgical System to perform a transvaginal hysterectomy in 2021. Based on the description from the developer, Memic Innovative Surgery, the Hominis system has a humanoid-shaped robotic arm with multi-planar flexibility and 360 degrees of articulation. Clinical studies gathered 30 hysterectomies performed by the Hominis system and showed that the transvaginal approach was completely successful without any device-related events or intraoperative complications [121].

#### 5.1.2. Radiotherapy and Chemotherapy

The killing of cancer cells can be achieved via the introduction of the high energy of X-rays or chemicals to ultimately shrink the tumour. Radiotherapy is executed where high dose energy, a range of 40 to 85 Grays (Gys) [122,123], depending on the size of tumour and the distance from adjacent normal tissue, is applied to the tumour. The standard protocol includes the combination of external-beam radiotherapy (EBRT) to the pelvic region and brachytherapy (BT) [124]. Brachytherapy is performed where a high dose of radiation is given directly to either within or adjacent to the tumour site to kill residual cancer cells at the primary tumour site. To reduce the adverse outcomes and effects on the organ adjacent to the uterine cervix, such as the rectum, sigmoid colon, and bladder, three-dimensional image-guided brachytherapy (3D-IGBT) using CT or MRI can efficiently deliver sufficient high doses of radiation to the target site [125,126,127]. Despite radiotherapy alone or the concurrent surgical removal of the tumour in practice [128], these primary treatments may not improve the overall survival of patients [129,130]. Radiotherapy improved the overall and cause-specific survival for patients at TNM stages III and IV, but may not be favourable for young patients with tumours of size <3 cm and at TNM stage I and II [131]. A combination of radiotherapy with chemotherapy may give a favourable clinical outcome.

Patients who received cisplatin mono-chemotherapy did not have improved overall survival [132]. The findings from clinical trials conducted two decades ago and recently consistently recommend the inclusion of concomitant cisplatin-based chemotherapy and radiotherapy or brachytherapy to treat patients with advanced cervical cancers [133,134,135,136,137]. Brachytherapy is often conducted to target a large tumour concomitant with or towards the end of chemotherapy [138]. In addition, a combination of cisplatin and another chemotherapeutic approach also provides a favourable outcome. Several clinical trials demonstrated that the patients who were diagnosed with advanced cervical carcinoma had a better progression-free survival (PFS) with lesser adverse reactions after receiving cisplatin in combination with 5-fluoracil (5-FU) [133,139], gemcitabine [136], ifosfamide [140,141], bleomycin [141], or paclitaxel [142] than those who were treated with hydroxyurea [133,137,139]. Conversely, in Japan, a Phase III trial on patients diagnosed with stage IB2, IIA2, or IIB cervical squamous carcinoma and treated with neoadjuvant chemotherapy (bleomycin, vincristine, mitomycin, or cisplatin) prior to radiotherapy did not improve the overall survival of the patients compared to those who received radiotherapy alone [143]. The trial was then terminated as the patients who received neoadjuvant chemotherapy did not show a better overall survival rate than those who underwent radiotherapy. More clinical trials should be conducted to inform the efficacy of the chemotherapeutics in treating cervical cancer patients of different cultural and ethnic backgrounds.

Despite the better treatment outcome, adverse events are inevitable. There are more patients who receive combinatorial treatments who suffer grade 3 or 4 toxicities than those who undergo mono-treatment. Treatment-related hematological, gastrointestinal, urological, and neurological toxicities, including neutropenia, leukopenia, thrombocytopenia, myelosuppression, gastrointestinal effects, pulmonary effects, cardiovascular effects, nausea, vomiting, diarrhoea, fatigue, alopecia, and weight loss are among the commonly reported side effects [133,136,139,140,141,142]. In addition, treatment-related death was also reported [136,142].

#### 5.1.3. Immune Checkpoint Inhibitors

Tumour cells exploit the immune checkpoint by expressing immunoreceptors on their cell surface, such as programme death 1 (PD-1) and cytotoxic T lymphocyte antigen 4 (CTL-4), allowing them to evade host immune surveillance. Immune checkpoint inhibitors work by blocking the binding of PD-1 to PD-1 ligand (PD-L1) or CTL-4 to cytotoxic T cells, thereby activating the T cells to recognise these tumour cells [143,144]. Immune checkpoint inhibitors, including the anti-PD1 (pembrolizumab, nivolumab, cemiplimab and balstilimab), anti-PDL1 (durvalumab) and anti-CTLA4 (ipilimumab, tremelimumab and zalifrelimab) monoclonal antibodies, have been effective in treating patients diagnosed with locally advanced, persistent, recurrent, and metastatic cervical cancer [145,146]. These treatment regimens are often given alone or in combination with chemotherapy.

Pembrolizumab has become a standard, safe, and effective treatment option for advanced cervical cancer. A Phase II trial, KEYNOTE-158 (NCT02628067), revealed that this PD-L1 inhibitor is safe and has produced manageable after-treatment effects [147], while in a Phase III clinical trial, KEYNOTE-862 (NCT036335567), patients with persistent, recurrent and metastatic cervical cancer received platinum-based chemotherapy with or without bevacizumab, and pembrolizumab prolonged the patients’ PFS and overall survival (OS) [148]. Other PD-1 inhibitors, such as nivolumab (NCT02257528) [149], atezolizumab (NCT03340376) [150], and cemiplimab (NCT03257267) [151], were also studied in Phase II or Phase III trials. Similar to pembrolizumab, treatment with durvalumab concurrent with platinum-based chemoradiotherapy also improved the PFS of patients with locally advanced cervical cancer (NCT03830866) [152].

Unlike pembrolizumab, ipilimumab monotherapy, an anti-CTL-4, showed modest efficacy in treating cervical squamous cell carcinoma (SCC) and adenocarcinoma [153]. Nonetheless, a combination of anti-PD-1 and CTL-4 could be a better option. In a Phase I/II Checkmate 385 study (NCT02488759), patients who received 1 mg/kg nivolumab with 3 mg/kg ipilimumab thrice weekly for four doses, followed by nivolumab maintenance twice weekly, had a longer PFS than the group who received 3 mg/kg nivolumab twice weekly with 1 mg/kg ipimumab six times weekly [154]. In addition, after receiving platinum-based chemotherapy, treatment with balstilimab and zalifrelimab (NCT03495882) showed a better objective response rate (ORR) than balstilimab alone (NCT03104699), for both cervical SCC and adenocarcinoma [155].

#### 5.1.4. Target-Specific Inhibitors

The overexpression of oncoproteins and kinases is often observed in various cancers. This makes them a good target for anti-cancer drug designing. For instance, under normal conditions, the expression of receptor tyrosine kinases (RTKs) is controlled at a low or undetectable level. However, in cancer cells, RTKs are upregulated, leading to the dysregulation of cell proliferation, growth, and migration. Several drugs targeting RTKs marched to clinical trials. The tolerability of patients for these drugs is generally good. Anlotinib is a novel drug developed by Chia-tai Tianqing Pharmaceutical Co., Ltd. (Lianyungang, China) that targets multiple RTKs, including vascular endothelial growth factors (VEGF1, VEGF2, and VEGF3), c-Kit, platelet-derived growth factor receptor-alpha (PDGFR-α), and the fibroblast growth factor receptors (FGFR1, FGFR2, and FGFR3) [156]. In a Phase I/II trial (NCT02558348), Anlotinib was well tolerated by cervical cancer patients [157]. However, the trial has been terminated by the sponsor.

Another prominent target for cancer treatment is the epidermal growth factor (EGRF). Monotherapy with anti-EGRF, gefitinib, and erlotinib, is well tolerated by patients. These drugs showed no ORR in advanced, recurrent, and metastatic cervical cancer [158,159]. However, when combining erlotinib with cisplatin-based chemoradiotherapy, the PFS and OS of patients with locally advanced cervical cancer were improved [160]. The clinical trials showed that patients who received anlotinib and erlotinib experienced grade 1 and 2 adverse events, including nausea, skin rash, diarrhoea, hypertension, oral pain, epistaxis, insomnia, headache, fatigue, anorexia, and urinary tract infection [157,158], while the majority of subjects who received gefitinib experienced grade 1 or 2 toxicities, and less than 10% of the subjects suffered grade 3 skin and gastrointestinal toxicities. No grade 4 toxicity was observed [159].

#### 5.1.5. Anti-Angiogenesis

In recent years, bevacizumab, a humanized monoclonal antibody that acts on neutralizing the vascular epidermal growth factor (VEGF), a key modulator involved in angiogenesis, has gained popularity. Phase II and III trials conducted by the Gynecologic Oncology Group (GOG) and the Spanish Research Group for Ovarian Cancer, revealed that bevacizumab combined with chemotherapy increased patients’ OS compared to chemotherapy alone [161,162]. Meanwhile, a Phase II trial (NCT03816553) revealed another selective VEGF 2 inhibitor, apatinib, which, when combined with camrelizumab, a fully humanized monoclonal antibody against PD-1, achieved a 55.6% ORR and an 8.8-month PFS in patients with advanced cervical cancer [163], compared to patients who received apatinib monotherapy (around 14–15% of ORR) [164,165].

The safety and efficacy of other anti-VEGF agents, including cediranib, pazopanib, and lapatinib were also explored. Compared to patients with metastatic or recurrent cervical cancer who were treated with carboplatin and paclitaxel, the addition of cediranib to these chemotherapeutics prolonged PFS, despite the increased toxicity [166]. Intriguingly, pazopanib monotherapy appears to exert a better anti-angiogenic and anti-tumour effect than lapatinib, with improved PFS. Later, a clinical trial was conducted to combine pazopanib and lapatinib. Unfortunately, this combination did not give a favourable treatment outcome and was discontinued as the futility boundary was crossed, and it had higher toxicity compared to the respective monotherapy [166].

#### 5.1.6. Drug-Antibody Conjugate

Tissue factor (TF) is a protein expressed abundantly in solid tumours, including cervical cancer. The aberrant expression of TF contributes to tumour growth, angiogenesis, metastasis, and thrombosis. Tisotumab vedotin is an investigational antibody-drug conjugate, which acts directly against TF. So far, tisotumab vedotin is the only drug-antibody conjugate that recently gained accelerated approval from the FDA. A Phase II trial (NCT03438396) revealed that tisotumab vedotin poses an antitumour activity, with 24% ORR and tolerable treatment-related toxicity [167]. This drug is currently undergoing a Phase III trial (innova TV 301, NCT04697628).

#### 5.1.7. HPV Vaccines

There are two types of vaccines designed for HPV-related diseases: HPV prophylactic and therapeutic vaccines. HPV prophylactic vaccines are essentially viral-like particles (VLPs) comprising the HPV L1 subunits. The HPV prophylactic vaccines gained FDA approval, and these vaccines have been included in HPV vaccination programmes worldwide. The 9-valent Gardasil^®^9 (protects against HPV6, 11, 16, 18, 31, 33, 45, 52, and 58) and quadrivalent Gardasil^®^4 (protects against HPV6, 11, 16, and 18) are produced by Merck (Kenilworth, NJ, USA), while bivalent Cervarix (protects against HPV16 and 18) is made by GlaxoSmithKline (Brentford, UK). Females aged 15 to 55 years old who received the AS04-HPV-16/18 vaccine (Cervarix) sustained 10-year immune protection, with anti-HPV16/18 titers higher than that of natural infection [168]. Whilst women who underwent surgical resection for HPV-related disease prior to receiving Gardasil^®^4 had a reduced risk of developing subsequent HPV-related disease, including HSIL (NCT00092521 and NCT00092534) [169]. Despite the efficacy of HPV prophylactic vaccines in preventing LSIL and HSIL of the uterine cervix, there is a lack of evidence as to whether or not the vaccines can provide immune protection against cervical cancer. Moreover, due to the increasing favouritism among the public over social media, the contradicting and somewhat misleading information poses a substantial influence on the public acceptance of HPV prophylactic vaccines [170]. This is undeniably a factor that adds to the challenge in the implementation of the HPV vaccination programme.

One important feature of HPV-associated malignancies is the abundant expression of the viral E6 and E7 oncoproteins, which are crucial elements for promoting and maintaining cancer phenotypes. In most cancers, the expression of other viral proteins might be disrupted. Hence, E6 and E7 are promising targets for the design of HPV therapeutic vaccines. The HPV therapeutic vaccines could treat persistent and recurrent HPV infections or HPV-associated malignancies. Ideally, these vaccines can elicit cell-mediated immunity to produce E6- and E7-specific CD4 and CD8 T cell responses, which may favour the regression of cervical lesions or cancer [171,172]. To date, there are various HPV therapeutic vaccines in clinical trials, including peptide-based, protein-based, DNA-based and DNA/RNA/bacterial-based vector recombinant vaccines.

Peptide-based HPV therapeutic vaccines are often combined with immunogenic adjuvant or added with agonist epitopes to elicit sufficient host immunological responses. A Phase II trial on a mix of nine HPV16E6 and four E7 synthetic long peptides (SLP) containing adjuvant Montanide ISA-51, showed that the treatment option can induce a broad interferon-gamma (INFγ)-associated T cell response in patients with advanced or recurrent gynaecological cancers, including cervical cancer, but did not induce cancer regression or prevent progression [173]. Another SLP in Phase I/II trials (NCT03821272, NCT02481414, NCT01653249), PepCan, consisting of four HPV16E6 synthetic peptides and adjuvant Candin^®^, is safe and effective in reducing viral load and increasing T-helper type 1 cells among women with high-grade cervical lesions [174,175]. Another HPV short peptide that marched to a clinical trial is the CIGB-228, which is an HLA-A2-restricted HPV16E7 peptide that was safe and able to induce IFNγ-associated T cell response, leading to the regression of high-grade lesions and HPV clearance [176]. Due to the positive treatment outcome, researchers are racing to produce effective peptide-based HPV vaccines. Other SLPs with known preclinical efficacy include Hespecta [177,178], SLP-CpG, which consists of an HPV16E7 SLP with a centrally located MHC I epitope [179], and NP-E7Lp, with E7 conjugated to ultra-small nanoparticles [180].

Protein-based vaccines are designed based on E6 and/or E7 proteins and are produced as fusion proteins. They often contain bacterial toxins and additional adjuvants, such as imiquimod [181], CpG or GI-0100 [182] to achieve recognition by antigen presenting cells (APCs) and to elicit cytotoxic T cell responses. The protein-based vaccines that marched to Phase I or II trials for cervical precancers are TVGV-1 (NCT02576561) [182], ProCervix (NCT01957878) [181], HSP-E7 or SGN-0010 (NCT00054041, NCT00091130) [183,184,185].

Unlike peptide- and protein-based therapeutic vaccines, viral (DNA or RNA) and bacterial vector vaccines are immunogenic and sufficient to elicit host rapid antibody and CD8 T cell responses. They can be easily engineered to carry immunogens of interest. One of the most used DNA virus vectors is of vaccinia origin, in which a large stretch of a gene of interest can be inserted into such a vector. For instance, the tipapkinogen sovacivec (TS) (NCT01022346) and TG4001 (NCT01022346) vaccines were produced from modified vaccinia virus Ankara (MVA), which is an attenuated and replication-deficient vector, carrying genes encoded for human cytokine IL-2, and modified forms of HPV16E6 and E7 proteins. Both of these vaccines were shown to be effective in reverting CIN2/3 histologic presentation, with viral clearance [186,187]. Another common DNA virus vector employed in vaccine development is the adenovirus vector. As adenovirus infection is common among the human population, Khan and colleagues constructed a replication-incompetent of a rare adenovirus type 26 recombined with HPV16 and 18 E2, E6, and E7 genes. The vaccine was able to spark a robust T cell response in the murine model [188]. Later, a Phase I/II was initiated to utilise an Ad26 vector carrying HPV16 and 18 immunogens as a prime immunisation, followed by MVA-HPV16/18 booster immunisation (NCT03610581). Unfortunately, the trial was terminated prematurely due to low enrolment and the COVID-19 pandemic. While the ADXS11-001 vaccine produced from live attenuated *Listeria monocytogenes*, which was engineered to produce full-length E7 encoded by HPV16 conjugated with listeriolysin-o (LLO), was in Phase II trials for recurrent and metastatic (CTRI/2010/091/001232 and NCT01266460) cervical cancers [189,190]. Treatment with ADXS11-001 alone or concurrent with cisplatin was comparable, with 12 months of 34–38% OS. A Phase III clinical trial for ADXS11-001 is ongoing and is expected to be completed in 2024.

In addition to these, the safety and efficacy of HPV DNA- and RNA-based vaccines were in trials for cervical precancerous lesions. One such vaccine is the VGX-3100, a DNA vaccine containing two plasmids of E6 and E7, encoded by HPV16 or 18. Intramuscular injection of VGX-3100 into patients with CIN2/3 was able to induce a robust cellular and humoral immune response, particularly in increasing interferon (IFN)γ and tumour necrosis factor (TNF)α production, as well as CD8+ T cell activation (NCT01304524, NCT01188850), leading to histological regression [191,192]. Meanwhile, DNA vaccines based on pNGVL4a plasmid-expressing HPV genes linked to either calreticulin (CRT) or *Mycobacterium tuberculosis* heat shock protein 70 (HSP70) were also developed. These vaccines are designated as pNGVL4a-CRT/E7 (NCT00988559) [193] and pNGVL4a-Sig/E7(detox)Hsp70 [194], respectively. Similarly, they can elicit robust host immune response and histopathologic regression. Intriguingly, a recent preclinical study suggested that the translational potential of pNGVL4a-Sig/E7(detox)Hsp70, boosted with tissue antigen HPV vaccinia virus-based vector HPV vaccine and PD-1 blockade monoclonal antibody, not only induces cytotoxic T cell responses but also extends the survival of mice [194].

On the other hand, the RNA-based vaccine is the emerging cutting-edge technology for the generation of safe and highly effective vaccines. The recent success is manifested by the vaccine for COVID-19, such as the BNT162b2 mRNA vaccine from Pfizer-BioNTech. For HPV malignancies, HPV16 RNA-LPX, where an E7 mRNA is encapsulated in RNA-lipoplex (LPX), is administered intravenously and selectively taken up by dendritic cells. In a mouse model, the vaccine possesses anti-tumour properties, induced robust E7-specific CD8 infiltration, and lasting memory response [194]. This vaccine is currently undergoing a Phase I clinical trial (HARE-40 trial; NCT03418480). As the HPV therapeutic vaccines are still in the early phases of clinical trials, besides pain at the injection site and/or fever, there is insufficient evidence to unleash the efficacy- and treatment-related toxicities of this emerging treatment modality.

## 6. Methods

### 6.1. Literature Search Strategies

The databases used for the search of primary literature were Pubmed, MEDLINE (EBSCOhost), Scorpus, and Google Scholar. To ensure all relevant influential factors were included in the search strategy, the MesH terms “cervical cancer”, “human papillomavirus and cervical cancer”, “pap smear”, “diagnosis of cervical cancer”, “human papillomavirus screening”, “computer-aided diagnosis for cervical cancer”, “robotic laparoscopy”, “cervical cancer treatment”, and “clinical trials for cervical cancer”, were used in the search engines.

### 6.2. Inclusion and Exclusion Criteria

We reviewed all the relevant accepted publications, with particular focus on screening, diagnosis, and treatment regimens for cervical cancer within the recent decade (2012 to 2022). We read the title, abstract, and full text of the article and chose eligible and appropriate articles to be included in this review. The topics included (1) the prevalence, incidence, and mortality of cervical cancer worldwide; (2) how HPV promotes cervical dysplasia and cervical cancer; (3) diagnostic and screening platforms, staging for cervical neoplasia and carcinoma, as well as treatment modalities in clinical trials for cervical cancer; (4) articles written in English; and (5) the accessibility of the full text of the articles. From the initial selection process, we assessed a total of 44,692 relevant articles. Inappropriate articles (n = 41,395) were excluded, and the remaining full text articles (n = 3297) were selected. After further stringent filtering with specific keywords, a total of 194 articles were included in this review article. Information from the selected articles was arranged, congregated, interpreted, and summarised.

## 7. Concluding Remarks

In conclusion, cervical cancer still poses a global health threat, particularly for women in underdeveloped and developing countries. An effective way to prevent and triage cervical cancer is the implementation of accurate screening strategies for the early detection of premalignant lesions of the cervix uteri. The primary screening strategy recommended for the triage of women with HPV-associated cervical lesions includes a combination of cytology, an HPV nucleic acid test, and the staining of biopsy tissues with p16 and/or Ki67 surrogate markers. Upon the detection of morphological anomalies, the presence of HPV nucleic acids and the expression of biomarkers in the tissue samples, these lesions and tumours are being classified based on standard guidelines. With the creation of state-of-the-art technology, artificial intelligence and robotic modules have been incorporated into diagnosis and treatment tools. These advancements provide a breakthrough in cancer treatment and aid in reducing the dependence on trained technologists and specialists, the workload, and the time from tissue processing to decision making. Despite the huge advantages, on top of the concern over cost-effectiveness, computer-assisted devices require further improvement as they are not without error. For instance, during diagnosis and surgical procedures, the flexibility and impromptu decision/judgement of surgeons and specialists are yet to be replaceable by the current technology. The downfall of relying on computer-aided tools is that the accountability of medical-legal issues should be taken into consideration, especially when diagnostic errors occur. Furthermore, practicality comes into question when implementing such a high-cost system in developing countries. Another concern faced by most low- and middle-income countries is the affordability of the HPV prophylactic vaccines. Even though HPV prophylactic vaccines successfully reduce cervical cancer incidence, the vaccine may have a higher protective value for females in their puberty or adolescent age who have not been exposed to HPV infection. Women who had been exposed to HPV infection, particularly the cancer-causing HPV genotypes, may have a chance to develop cervical cancer. To date, mono-treatment, either through surgical resection, radiotherapy, chemotherapy, immunotherapy, and anti-cancer drugs may not be effective in curing cervical cancer. Clinical trials revealed that a combination of these treatments could improve progression-free survival and the overall survival of patients with advanced, recurrent, and metastatic cervical cancer. In addition to cervical cancer, these treatment modalities are also applicable to HPV-related cancers. Although the clinical practice guidelines for the prevention and treatment of cervical cancer are published, the implementation of standardised and well-defined treatment regimens could be challenging. Moreover, the field is still in urgent need of precision medicine, where a safe, effective, and target-specific drug that acts on HPV-containing cervical cancer cells is needed.

## Figures and Tables

**Figure 1 cancers-14-02913-f001:**
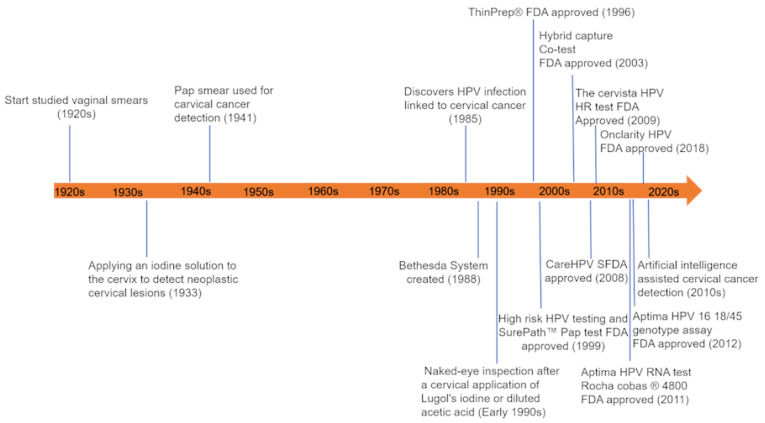
The schematic summarizes the timeline of the development of cervical cancer detection. The study of vaginal smears began in the 1920s, followed by the cytologic identification of neoplastic lesions using iodine solution (1930s) and the Papanicolaou stain (1941). After the discovery of the causal link between HPV infection and cervical cancer in 1985, HPV nucleic acid tests were included as a standard diagnostic test in the laboratory. In recent years, the artificial intelligence (AI)-assisted detection platform has become popular [21,22,23,24,25,26,27,28,29,30,31,32,33].

**Figure 2 cancers-14-02913-f002:**
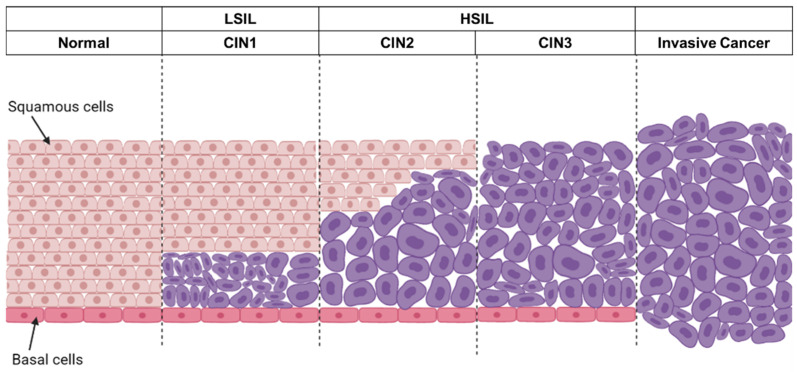
An overview of the progression of cervical cancer. Squamous cells are found in the outer part of the cervix, where a single layer of basal cells is attached to the basement membrane. Under persistent HR-HPV infection, cervical intraepithelial neoplasia may progress to invasive cancer. For cervical intraepithelial neoplasia 1 (CIN1) or low-grade squamous intraepithelial lesion (LSIL), dysplasia occurred in the lower 1/3 or less of the epithelium. CIN2/3, also called high-grade squamous intraepithelial lesion (HSIL), had significantly more dysplasia.

**Figure 3 cancers-14-02913-f003:**
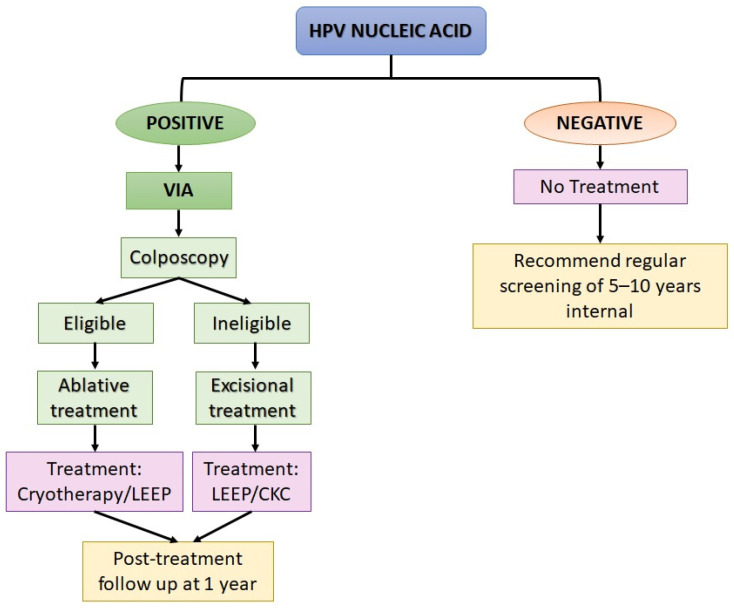
A summary of screen-and-treat strategies recommended by the World Health Organization (WHO) for screening and treatment of precancerous lesions for cervical cancer prevention [103]. LEEP, Loop electrosurgical excision procedure; CKC, cold knife conization.

**Table 1 cancers-14-02913-t001:** TNM classification 8th edition and 2018 FIGO Staging System for Uterine Cervical Cancer.

TNM	FIGO	Description
TI	I	Carcinoma is strictly confined to the cervix (extension to the uterine corpus should be disregarded)
TIA	IA	Invasive carcinoma that can be diagnosed only with microscopy, with maximum depth of invasion < 5 mm
TIA1	IA1	Stromal invasion < 3 mm in depth
TIA2	IA2	Stromal invasion ≤ 3 mm and <5 mm in depth
TIB	IB	Invasive carcinoma confined to the uterine cervix, with measured deepest invasion ≥ 5 mm
TIB1	IB1	Tumor measures < 2 cm in greatest dimension
TIB2	IB2	Tumor measures ≤ 2 cm and < 4 cm in greatest dimension
TIB3	IB3	Tumor measures ≥ 4 cm in greatest dimension
TII	II	Carcinoma invades beyond the uterus, but has not extended onto the lower third of the vagina or to the pelvic wall
TIIA	IIA	Limited to the upper two-thirds of the vagina without parametrial involvement
TIIA1	IIA1	Tumor measures < 4 cm in greatest dimension
TIIA2	IIA2	Tumor measures ≥ 4 cm in greatest dimension
TIIB	IIB	With parametrial involvement but not up to the pelvic wall
TIII	III	Carcinoma involves the lower third of the vagina and/or extends to the pelvic wall and/or causes hydronephrosis or nonfunctioning kidney and/or involves pelvic and/or para-aortic lymph nodes
TIIIA	IIIA	Involves the lower third of the vagina, with no extension to the pelvic wall
TIIIB	IIIB	Extension to the pelvic wall and/or hydronephrosis or nonfunctioning kidney from tumor
TIIIC	IIIC	Involvement of pelvic and/or para-aortic lymph nodes, irrespective of tumor size and extent
TIIIC1	IIIC1	Pelvic lymph node metastasis only
TIIIC2	IIIC2	Para-aortic lymph node metastasis
TIV	IV	Carcinoma has extended beyond the true pelvis or has involved (biopsy-proven) the mucosa of the bladder or rectum
TIVA	IVA	Spread to adjacent pelvic organs
TIVB	IVB	Spread to distant organs

**Table 2 cancers-14-02913-t002:** Regional Lymph Nodes classification from TNM (8th Edition) classification system for cervical cancer.

NX	Regional lymph nodes cannot be assessed
N0	No regional lymph node metastasis
N1	Regional lymph node metastasis

**Table 3 cancers-14-02913-t003:** Distant Metastasis from TNM (8th Edition) classification system for cervical cancer.

MX	Distant metastasis cannot be assessed
M0	No distant metastasis
M1	Distant metastasis

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
