# Peer review of "Review of the Standard and Advanced Screening, Staging Systems and Treatment Modalities for Cervical Cancer"

_cancers, 2022, doi:10.3390/cancers14122913_

Round 1

Reviewer 1 Report

The paper is not balanced, some topics are described very superficially (first part), and others in too much detail (vaccines, esp. therapeutic). In biomarkers - cytological dual staining should be mentioned. In classification of intraepithelial lesions recent WHO classification for histology should be mentioned. The Figure 2 is not correct for CIN2. Screen and treat strategies are described wrongly, just like the Figure 3.  In therapy - the surgery for invasive cancer is missing, together with sentinel LN application. 

Author Response

Comments and suggestions for Authors: The paper is not balanced, some topics are described very superficially (first part), and others in too much detail (vaccines, esp. therapeutic). In biomarkers - cytological dual staining should be mentioned. In classification of intraepithelial lesions recent WHO classification for histology should be mentioned. The Figure 2 is not correct for CIN2. Screen and treat strategies are described wrongly, just like the Figure 3.  In therapy - the surgery for invasive cancer is missing, together with sentinel LN application. 

Respond to Reviewer’s comments:

We appreciate the valuable comments from the reviewer, which help us to improve quality and clarity of our manuscript tremendously. We agreed with the reviewer’s comments, and made the following amendments as recommended:

  • With regards to the first part of the article, we agreed that it was superficial and not written in great details. We think that these aspects of HPV, e.g. background, epidemiology and viral life/infectious cycle have been covered in great details in numerous review articles. In view of that, we focus more on other aspects of the HPV and cervical abnormalities, especially the updates on diagnosis, detection and treatment options. As for the HPV prophylactic vaccine and treatments, including surgery, radiotherapy, chemotherapy, immunotherapy, anti-angiogenic and anti-tumour drugs were included in clinical practice guidelines and discussed in previous review articles. We think it is worth including these in the current review article. While the other these areas that are emerging and advancing in the recent years, like the incorporation of computer-aided technologies and different types of HPV therapeutic vaccines in clinical trials, were also discussed in this article.

  • We added the following sentence under Section 3.1.4, Line 223 – 225: “Dual staining of these biomarkers, for instance, p16 and p53 immunohistochemistry test has been commonly performed in many laboratories worldwide.”

  • We added the following sentences under Section 1, Line 47 - 61, to describe the WHO histologic classification on tumours of the uterine cervix: “In general, the WHO defined premalignant lesions of the uterine cervix as cervical intraepithelial neoplasia (CIN)[10]. While the epithelial tumours of the uterine cervix are histologically classified into squamous cell carcinoma, glandular tumours and precursors, mesenchymal tumours and tumour-like conditions, mixed epithelial and mesenchymal tumours, melanocytic tumours, miscellaneous tumours, lymphoid and haematopoietic tumours, and secondary tumours[11]. Among these, squamous cell carcinoma (75%) is the most commonly reported cervical cancer type, followed by adenocarcinoma, a type of glandular tumours and precursors (25%)[12,13]. HPV16 is predominantly found in squamous cell carcinoma, while HPV18 is more commonly detected in adenocarcinomas and adenosquamous carcinoma[14]. For these 2 types of cervical cancers, the risk factors, HPV genotypes detected and treatment are similar[15]. While for the usual endocervical adenocarcinoma types, such as mucinous, micropapillary and villoglandular are associated with HPV, while gastric, clear cell, endometrioid and mesonephric types are often not associated with HPV[15,16].”

  • Figure 2 has been revised and amended.

  • Under Section 5.1, description for Figure 3 has been revised and amended, as follow:
  • Line 384: “via colposcopy” was added.
  • Line 384 – 385: “in the case where HPV nucleic acid is detected from the tissue samples but” was deleted.
  • Line 387: “via colposcopy” was added.
  • Line 388 – 389: “Rescreening within a year is recommended for HPV infected patients with precancerous lesions observed.” was deleted.
  • Line 390 – 391: “The patients should attend follow up within a year after the treatment.” was added.
  • Under Section 5.1, Line 400 – 410, the following paragraph was added to include surgery and lymphadenectomy:

“For patients diagnosed with precancer or malignant cervical cancer, surgery is the main therapeutic measure. The precancerous lesions can be removed via cryosurgery, laser ablation and conization. For the treatment of cervical cancer, the National Comprehensive Cancer Network (NCCN)[109] and European Society for Medical Oncology (EMSO) [110] published comprehensive guidelines. For advanced, invasive and metastatic cervical cancer, the tumour is usually removed by performing hysterectomy (simple or radical), trachelectomy[111] and pelvic exenteration[112], depending on the size and nature of the tumour. Often, when performing radical hysterectomy, para-aortic lymph node sampling, pelvic lymphadenectomy, or sentinel lymph node mapping and biopsy is done[111]. It is crucial to know the involvement of lymph nodes for the planning of the subsequent adjuvant therapy.”

Reviewer 2 Report

This review summarizes the screening, diagnosis, and treatment for cervical cancer.  This is an excellent review and very well written. I have only a few minor points the authors might consider incorporating in the final form.

  1. This review emphasizes HPV. About 10% of cervical carcinomas are not HPV-related. The authors may want to touch on this point. The authors can also briefly discuss endocervical adenocarcinoma regardless of HPV status.
  2. How is HPV-related cervical squamous cell carcinoma different from HPV-related squamous cell carcinoma in terms of treatment?
  3. Medical-legal issue of relying on computer-aided technology: as the authors stated, this technology requires oversight from trained professionals who are responsible for the patients’ test result. When diagnostic errors occur, a person or an organization will be held accountable legally.
  4. In section 4.1 the authors may add: many cases of CIN1/LSIL may not be related to high risk HPV.
  5. Line 514: any reason why this trial was terminated?
  6. The authors can discuss very briefly the application of next generation sequencing on cervical dysplasia and invasive carcinoma, including liquid biopsy to detect and monitor carcinoma cells/nucleic acid in plasma.
  7. Line 47: change “Hpv’ to “HPV”

Line 182: add a comma after “Qiagen”

Line 183: I thought that Hologic closed Madison branch. Could you please check if Cervista is still associated with Madison, Wisconsin?

Line 353: please spell out for “VIA” the first time introducing it.

Line 464: change “1B2” to “IB2”

Line 558: add “)” after “UK”

Author Response

Comments and Suggestions for Authors: This review summarizes the screening, diagnosis, and treatment for cervical cancer.  This is an excellent review and very well written. I have only a few minor points the authors might consider incorporating in the final form.

Respond to Reviewer’s comments:

We appreciate that the reviewer spent time to carefully read our article, and give us constructive and valuable comments. This indeed helps us to improve quality and clarity of our manuscript tremendously. We agreed with the comments from the reviewer, and made the following amendments as recommended (in blue):

  1. This review emphasizes HPV. About 10% of cervical carcinomas are not HPV-related. The authors may want to touch on this point. The authors can also briefly discuss endocervical adenocarcinoma regardless of HPV status.
  • Based on reports from Walboomers JM and colleagues (Walboomers JM, et al., Journal of Pathology, 1999. 189(1):12-19), the study retested the HPV-positive and -negative samples. The study showed that HPV prevalence in cervical carcinoma is 99.7% worldwide. HPV DNA was not detected in the vast majority of HPV-negative cervical cancer specimens was due to technical limitation, for instance sampling, undetectable HPV gene via PCR due to disruption or low copy number of the target viral gene(s).

  • Section 1, Line 47- 61, a paragraph was added to describe the WHO histologic classification on tumours of the uterine cervix: “In general, the WHO defined premalignant lesions of the uterine cervix as cervical intraepithelial neoplasia (CIN)[10]. While the epithelial tumours of the uterine cervix are histologically classified into squamous cell carcinoma, glandular tumours and precursors, mesenchymal tumours and tumour-like conditions, mixed epithelial and mesenchymal tumours, melanocytic tumours, miscellaneous tumours, lymphoid and haematopoietic tumours, and secondary tumours[11]. Among these, squamous cell carcinoma (75%) is the most commonly reported cervical cancer type, followed by adenocarcinoma, a type of glandular tumours and precursors (25%)[12,13]. HPV16 is predominantly found in squamous cell carcinoma, while HPV18 is more commonly detected in adenocarcinomas and adenosquamous carcinoma[14]. For these 2 types of cervical cancers, the risk factors, HPV genotypes detected and treatment are similar[15]. While for the usual endocervical adenocarcinoma types, such as mucinous, micropapillary and villoglandular are associated with HPV, while gastric, clear cell, endometrioid and mesonephric types are often not associated with HPV[15,16].”

  1. How is HPV-related cervical squamous cell carcinoma different from HPV-related squamous cell carcinoma in terms of treatment?
  • We added a sentence under the Section 7, Line 770 – 771: “Besides cervical cancer, these treatment modalities are applicable also applicable to HPV-related cancers.”

  1. Medical-legal issue of relying on computer-aided technology: as the authors stated, this technology requires oversight from trained professionals who are responsible for the patients’ test result. When diagnostic errors occur, a person or an organization will be held accountable legally.
  • We completely agree with the reviewer. We added the following sentence under Conclusion, Line 756 – 758:

“The downfall of relying on computer-aided diagnostic tools is that, the accountability of medical legal issues should be taken into consideration, especially when diagnostic errors occur.”

  1. In section 4.1 the authors may add: many cases of CIN1/LSIL may not be related to high risk HPV.
  • The sentence “Many cases of CIN1 or LSIL may not be related to high-risk HPV genotypes.” had been added to Line 280 – 281.

  1. Line 514: any reason why this trial was terminated?
  • In regards to clinical trial for Anlotinib, we were too curious of the reason. However, we couldn’t find an explanation to it. The reason was neither mentioned in the clinical trial record (ClinicalTrials.gov) or the relevant research article (Werner et al., 2017).

  • As for another Phase III trial by the Japan Clinical Oncology Group trial, we added the reason of the trial terminated as follow:

Section 5.1.2, Line 511 – 513: “as the patients who received neoadjuvant chemotherapy did not show a better overall survival rate than those who underwent radiotherapy.”

  1. The authors can discuss very briefly the application of next generation sequencing on cervical dysplasia and invasive carcinoma, including liquid biopsy to detect and monitor carcinoma cells/nucleic acid in plasma.
  • Under Section 3.1.3, Line 210 – 216, the following sentences were added about using NGS:

“Furthermore, the next generation sequencing (NGS) can be explicitly used to improve the specificity and sensitivity of HPV detection. The NGS panel can effectively detect the presence of all HPV genotypes presence in archival formalin-fixed, paraffin-embedded (FFPE), liquid-based cytology (LBC) and plasma samples[62,63]. Thus, the NGS technology lower the chance of false-negative results compared with traditional PCR-based assays, and provides more accurate screening results for the subsequent diagnosis and treatment planning.”

  1. Line 47: change “Hpv’ to “HPV”
  • “Hpv” had been amended to “HPV”

  1. Line 182: add a comma after “Qiagen”
  • A comma had been added after Qiagen (Line 196).

  1. Line 183: I thought that Hologic closed Madison branch. Could you please check if Cervista is still associated with Madison, Wisconsin?
  • Indeed, Hologic closed Madison branch. We changed it to CervistaTM (Hologic, San Diego, CA, USA) (Line 197), as well as APTIM HPV assay (Hologic, San Diego, CA, USA) (Line 200).

  1. Line 353: please spell out for “VIA” the first time introducing it.
  • Line 377: “VIA” had been spelled out as visual inspection with acetic acid (VIA)

  1. Line 464: change “1B2” to “IB2”
  • Line 507: The typo “1B2” had been changed to “IB2”.

  1. Line 558: add “)” after “UK”
  • Line 618: “)” had been added after UK.

Reviewer 3 Report

This review paper, entitled Review of the standard and advanced screening, staging systems and treatment modalities for cervical cancer, was very excellent. I suggest accept in present form. 

The title changed to be “State of the art in cervical cancer screening and treatment modalities’. Redundant references should be removed. When treatment is discussed, the NCCN guidelines for cervical cancer or ESMO clinical practice guidelines for cervical cancer have been widely announced. What treatment strategy is authors’ opinion? Advanced treatment modalities, including complications and contraindications in clinical practice or in clinical trial, should be described detailed.

Author Response

Comments and suggestions for Authors: This review paper, entitled Review of the standard and advanced screening, staging systems and treatment modalities for cervical cancer, was very excellent. I suggest accept in present form. The title changed to be “State of the art in cervical cancer screening and treatment modalities. Redundant references should be removed. When treatment is discussed, the NCCN guidelines for cervical cancer or ESMO clinical practice guidelines for cervical cancer have been widely announced. What treatment strategy is authors’ opinion? Advanced treatment modalities, including complications and contraindications in clinical practice or in clinical trial, should be described detailed.

Respond to Reviewer’s comments:

We appreciate that the reviewer spent time to carefully read our article, and give us constructive and valuable comments. This indeed helps us to improve quality and clarity of our manuscript tremendously. We agreed with the comments from the reviewer, and made the following amendments as recommended (in blue):

  • References had been checked, updated and amended.

  • We included the following paragraph to mentioned the NCCN and ESMO guidelines for cervical cancer:

Under Section 5.1, Line 399 – 409, the following paragraph was added to include surgery and lymphadenectomy:

“For patients diagnosed with precancer or malignant cervical cancer, surgery is the main therapeutic measure. The precancerous lesions can be removed via cryosurgery, laser ablation and conization. For the treatment of cervical cancer, the National Comprehensive Cancer Network (NCCN)[108] and European Society for Medical Oncology (EMSO) [109] published comprehensive guidelines. For advanced, invasive and metastatic cervical cancer, the tumour is usually removed by performing hysterectomy (simple or radical), trachelectomy[110] and pelvic exenteration[111], depending on the size and nature of the tumour. Often, when performing radical hysterectomy, para-aortic lymph node sampling, pelvic lymphadenectomy, or sentinel lymph node mapping and biopsy is done[110]. It is crucial to know the involvement of lymph nodes for the planning of the subsequent adjuvant therapy.”

  • We agreed with the reviewer. We added descriptions about the adverse events, contradictions and complications in clinical practices and in clinical trials:

  • As described under Section 7, Line 764 – 770, we think that monotherapy may not be effective. For advanced, invasive and metastatic tumour, surgical resection and lymphadenectomy could be first performed. Based on the involvement of lymph node and distant metastasis, as there are more clinical trials with solid observations, a combination of 2 chemotherapeutics, or concomitant radiotherapy/chemotherapy or chemotherapy/immunotherapy could be suggested to patients. We also added to mention that other than cervical cancer, these treatment modalities are also applicable to HPV-related cancers.

  • For robotic-assisted laproscopy, we previously mentioned the challenges of this technology, as described in Line 446 – 448: “One major disadvantage of the master-slave robot is that the master console requires large space and high installation and operating costs [119,120].”

  • We also added the following descriptions on adverse events, complications and contradictions on the treatment modalities:
  • Section 5.1.1, Line 465 – 470: “The ergonomic design of the instrument can be improved; however, the installation and operating cost remain a major concern. The average cost for robotic-assisted laproscopy is significantly costlier than laproscopic surgery, which was estimated to be US$12,340 ± $5880 and US$10,227 ± $4986, respectively. This higher cost is predominantly related to the operating procedure[124].”

Under Section 5.1.2, Line 515 – 522: “Despite the better treatment outcome, the adverse events are inevitable. There are more patients who received combinatorial treatments suffered grade 3 or 4 toxicities than those who undergo mono-treatment. Treatment-related hematological, gastrointestinal, urological and neurological toxicities, including neutropenia, leukopenia, thrombocytopenia, myelosuppression, gastrointestinal, pulmonary, cardiovascular, nausea, vomiting, diarrhoea, fatigue, alopecia and weight loss are among the commonly reported side effects [137,140,143–145,147]. In addition, treatment-related death was also reported[140,147].

  • Section 5.3.4, Line 573 – 578: “The clinical trials showed that patients who received anlotinib and erlotinib experienced grade 1 and 2 adverse events, including nausea, skin rash, diarrhoea, hypertension, oral pain, epistaxis, insomnia, headache, fatigue, anorexia and urinary tract infection [162,163]. While majority of subjects who received gefitinib experienced grade 1 or 2 toxicities, and less than 10% of subjects suffered grade 3 skin and gastrointestinal toxicities. No grade 4 toxicity was observed[164].”

  • Section 5.3.6, Line 624 – 628: “Moreover, due to the increasing favoritism among the public over social media, the contradicting and somewhat misleading information poses a substantial influence on the public acceptance to receiving HPV prophylactic vaccines[124]. This is undeniably a factor that add to the challenge in the implementation of HPV vaccination programme.”

  • Section 5.3.6, Line 709 – 712: “As the HPV therapeutic vaccines are in still early phases of clinical trials, besides pain at injection site and/or fever, there are insufficient evidence to unleash the efficacy and treatment-related toxicities about this emerging treatment modality.”

  • Section 7, Line 751 – 757 (added part were underlined): Despite the huge advantages, on top of the concern over cost-effectiveness, computer-assisted devices require further improvement as they are not errorless-prone. For instance, during diagnosis and surgical procedures, flexibility and impromptu decision/judgement of surgeons and specialists are yet-to-be replaceable by the current technology. The downfall of relying on computer-aided diagnostic tools is that, the accountability of medical legal issues should be taken into consideration, especially when diagnostic errors occur.

  • Under Section 7, Line 769 – 772: “Besides cervical cancer, these treatment modalities are also applicable to HPV-related cancers. Although the clinical practice guidelines for the prevention and treatment of cervical cancer are published, the implementation of a standardised and well-defined treatment regimens could be challenging.”

Round 2

Reviewer 1 Report

Dera authors,

please correct the chapter: 5.1. Screen-and-treat strategies

You have mixed screen and treat with screen, triage and treat. Please use:

https://www.who.int/publications/i/item/9789240030824

Author Response

We would like to express our utmost appreciation to the reviewer for the time spent on further reviewing and pointing out the lack of our manuscript. We agree that the screen-and-treat strategies should be corrected, as well as Figure 3, as shown below:

Under Section 5.1, Line 371 – 392:

“The screen-and-treat approach is mainly recommended for women of age >30, due to their higher cervical cancer risk than younger women. The screen-and-treat strategy is summarised in Figure 3. According to the guidelines published by the WHO[103], the HPV nucleic acid detection and visual inspection with acetic acid (VIA) are the primary screening tests. It is strongly recommended for women of age 30 to 49 years old should perform regular screening of 5 to 10 years intervals. While for women of age 50 and above, after receiving 2 consecutive negative cervical cancer screening results, further screenings is not necessary.

To perform HPV nucleic acid detection test, the samples can either be self-sampled or collected by a health practitioner. In the case when HPV DNA is detected, depending on the eligibility, women should be assessed through visual inspection with acetic acid (VIA). VIA at the transformation zone is recommended for pre-menopausal women. This is because the transformation zone starts to recede during menopause. Ablative treatment can be performed via cryotherapy, in which the abnormal lesions is removed by heating it with thermal coagulation. Meanwhile for high-grade lesions (CIN 2 or 3), cryotherapy is the first choice of treatment, provided that the entire lesion and squamous-columnar junction are visible, and the lesion does not cover 75% of the ectocervix region. In the case when cryotherapy is not recommended, excisional treatment via loop electrosurgical excision procedure (LEEP) can be performed[102]. For women who are ineligible to perform ablative treatment, LEEP or cold knife conization should be performed, followed by histological assessment for lesion classification. The patients should attend follow up within a year after the treatment.”